# In-Situ Electrochemical Exfoliation and Methylation of Black Phosphorus into Functionalized Phosphorene Nanosheets

**DOI:** 10.3390/ijms24043095

**Published:** 2023-02-04

**Authors:** Aidar M. Kuchkaev, Airat M. Kuchkaev, Aleksander V. Sukhov, Svetlana V. Saparina, Oleg I. Gnezdilov, Alexander E. Klimovitskii, Sufia A. Ziganshina, Irek R. Nizameev, Igor P. Asanov, Konstantin A. Brylev, Oleg G. Sinyashin, Dmitry G. Yakhvarov

**Affiliations:** 1Arbuzov Institute of Organic and Physical Chemistry, FRC Kazan Scientific Center of RAS, Arbuzov Street 8, 420088 Kazan, Russia; 2Alexander Butlerov Institute of Chemistry, Kazan Federal University, Kremlyovskaya Street 18, 420008 Kazan, Russia; 3Institute of Physics, Kazan Federal University, Kremlyovskaya Street 18, 420008 Kazan, Russia; 4Zavoisky Physical-Technical Institute, FRC Kazan Scientific Center of RAS, Sibirsky Tract 10/7, 420029 Kazan, Russia; 5Nikolaev Institute of Inorganic Chemistry, Siberian Branch of the Russian Academy of Sciences, 3 Academician Lavrentiev Avenue, 630090 Novosibirsk, Russia

**Keywords:** black phosphorus, phosphorene, covalent functionalization, electrochemical exfoliation

## Abstract

Two-dimensional black phosphorus (BP) has attracted great attention as a perspective material for various applications. The chemical functionalization of BP is an important pathway for the preparation of materials with improved stability and enhanced intrinsic electronic properties. Currently, most of the methods for BP functionalization with organic substrates require either the use of low-stable precursors of highly reactive intermediates or the use of difficult-to-manufacture and flammable BP intercalates. Herein we report a facile route for simultaneous electrochemical exfoliation and methylation of BP. Conducting the cathodic exfoliation of BP in the presence of iodomethane makes it possible to generate highly active methyl radicals, which readily react with the electrode’s surface yielding the functionalized material. The covalent functionalization of BP nanosheets with the P–C bond formation has been proven by various microscopic and spectroscopic methods. The functionalization degree estimated by solid-state ^31^P NMR spectroscopy analysis reached 9.7%.

## 1. Introduction

In recent years, two-dimensional (2D) black phosphorus (BP) has gained notable attention in the scientific community due to its unique in-plane anisotropic structure, high carrier mobility, adjustable direct bandgap and other outstanding physical and chemical properties, which find applications in various micro- and optoelectronic, energy and sensing devices, catalytic systems and in biomedicine [1,2,3,4,5,6]. However, BP and especially its exfoliated form—the phosphorene nanosheets—suffer from poor environmental stability: the material is readily oxidized in the presence of air and moisture, which hinders its processing in various ambient conditions [7,8,9]. Chemical functionalization of BP is a great tool for enhancing a material’s ambient stability, dispersibility, its intrinsic electronic, optic or catalytic properties and is a promising strategy for the preparation of new BP-based functional materials [8,9,10]. Currently, covalent functionalization of BP with high-reactive intermediates (carbon free radicals or nitrenes) using diazonium or iodonium salts or organic azides as functionalization agents has been extensively explored [9,11,12,13,14]. In addition, some other successful approaches for covalent functionalization with organic substrates, such as nucleophilic compounds, are known [15].

The use of electrophilic agents, such as alkyl halides, for the functionalization of BP is challenging due to the low reactivity of BP toward these compounds. However, the introduction of a partial negative charge on the surface of BP makes it possible to significantly increase the activity of the material in this reaction [16,17,18]. Thus, Hirsch et al. have developed a reductive alkylation route [17], which had been previously implemented for the functionalization of carbon materials [19,20,21,22,23]. The method consists of the preparation of reduced forms of BP via the intercalation of alkali metals between the BP layers. The resulting intercalates are able to react with alkyl halides, yielding the alkylated material. However, it should be noted that this method is difficult to implement, as it requires extreme caution and highly inert conditions due to the explosive nature of BP intercalates.

In this work, we demonstrate a simple and efficient way for in-situ electrochemical exfoliation and methylation of BP. The functionalization of the material is proceeded by using BP as a cathode in electrochemical exfoliation conditions in the presence of iodomethane CH_3_I.

## 2. Results and Discussion

The possibility of simultaneous BP exfoliation and CH_3_I reduction was investigated using the cyclic voltammetry (CV) method. The CV curves of a 0.01 M Bu_4_NBF_4_ solution in DMSO for Pt and BP electrodes are shown in Figure 1a. The data obtained for the Pt electrode implies that the process of supporting electrolyte decomposition begins at the potential of −2.7 V. In the case of the BP electrode, a noticeable current increase is observed, which begins at approximately −0.75 V. Such a broadening of the signal is occurred due to the process of intercalation of tetrabutylammonium ions between the BP layers, which correlates well with the literature data describing the mechanism of cathodic exfoliation of BP [24]. Further potential sweep to −3 V leads to the decomposition of both solvent and supporting electrolyte with the release of gaseous compounds, which are responsible for the exfoliation of BP.

After that, the CV curves for the BP cathode were recorded under the same conditions but in the presence of increasing amounts of CH_3_I (Figure 1b). The addition of CH_3_I to the working solution does not affect the process of intercalation of tetrabutylammonium ions between layers of BP, as can be seen from the maintaining of the CV curve morphology upon potential sweep up to −2.0 V. The current increase upon further cathodic potential sweep is associated with the reduction of CH_3_I molecules with the formation of methyl radicals [25], followed by the reduction of Bu_4_NBF_4_. Thus, the conducted CV experiments prove the possibility of the simultaneous CH_3_I reduction reaction along with the intercalation of tetraalkylammonium cations with their subsequent reduction. It should also be noted that the reduction of CH_3_I molecules proceeds at fewer cathodic potentials compared to the reduction of supporting electrolytes; therefore, this process will proceed primarily at applied high negative potential.

Next, the electrochemical exfoliation of BP in a 0.01 M DMSO solution of Bu_4_NBF_4_ in the presence of 0.3 M CH_3_I was performed. To neutralize the influence of I_2_ and other oxidation products that are generated during the exfoliation/functionalization process, the experiment was conducted in a cell with the separation of the cathode and anode spaces (Figure 2). The process was carried out with an applied voltage of −20 V for 25 min. The obtained functionalized material (BP–CH_3_) was then collected by centrifugation, washed with DMSO and several portions of isopropanol, and dried in vacuo for further characterization.

The size and thickness of the resulting BP–CH_3_ nanosheets were determined by the atomic force microscopy (AFM) and transmission electron microscopy (TEM) methods. As shown in the AFM and TEM images (Figure 3), the BP–CH_3_ sample exhibits a few-layer sheet-like morphology with well-defined edges and uniform thickness, which indicates the well-maintained crystallinity of the functionalized material. Analysis of the obtained data revealed that BP nanosheets with lateral size of 0.8–1.0 µm and a height of 8–10 nm (14–18 phosphorene layers) are formed during the electrochemical exfoliation process.

Further, a variety of spectroscopic methods were implemented to characterize the structure and composition of the functionalized material. The ^31^P magic angle spinning (MAS) NMR spectrum for BP exhibits an intense peak centered at 17.96 ppm, which corresponds to the characteristic signal of the BP lattice (Figure 4) [26]. The ^31^P MAS NMR spectrum of the BP–CH_3_ sample reveals a slight shift of the main BP signal (δ = 16.84 ppm), indicating partial distortion of the BP structure and an increase in the bandgap [13]. Functionalization of the BP nanosheets is associated with the appearance of a broad shoulder in the range of ca. 13 ppm to −40 ppm [13,14,27]. The signal at −0.20 ppm corresponds to phosphoric acid, which is formed as a result of partial oxidation of BP during exfoliation and product isolation; the broadened peak at 7.30 ppm refers to a series of overlapped signals of various phosphoric acids [28], as well as methyl phosphate, which can be formed by the reaction of H_3_PO_4_ with CH_3_I. The broadened signals at −18 and −32 ppm are of most interest. The analysis of the literature data has shown that the signals in this region correspond to the P–CH_3_ groups in organophosphorus compounds, for example, in 1-methylphospholane (δ= −32 ppm) or methyldiphenylphosphine (δ= −28 ppm) [29,30,31]. In addition, similar signals in the ^31^P NMR spectrum are also observed for some BP-based materials, in which the formation of the P–C covalent bond is realized [14,32]. Moreover, the ^1^H→^31^P cross-polarization magic angle spinning (CPMAS) NMR spectrum of BP–CH_3_ demonstrates an increase in the intensity of signals at −18 and −32 ppm indicating the presence of H atoms at a distance of one or two bonds from the corresponding phosphorus atoms. These facts strongly support the functionalization of BP with methyl groups. The emergence of two signals in the CPMAS NMR spectrum may be due to the different functionalization of phosphorus atoms by methyl groups. We assume that the signal at −18 ppm refers to the phosphorus atoms each bound with one methyl group, while the signal at −32 ppm is attributed to the phosphorus atoms bound with two methyl groups with the formation of the P(CH_3_)_2_ fragment. It is worth noticing that a similar trend towards the upfield shift of signals in ^31^P NMR spectra was observed for tertiary organic phosphines with different numbers of alkyl groups bound to the phosphorus atom (e.g., δ(Ph_2_PCH_3_) = −28 ppm, δ(PhP(CH_3_)_2_) = −46 ppm, δ(P(CH_3_)_3_) = −62 ppm [30]). Additionally, a comparison of the integral intensities of the signals at −18 and −32 ppm recorded in MAS and CPMAS modes showed that the ratio of the areas of the deconvoluted peaks S_−18ppm_:S_−32pmm_ changed from 1:0.38 in the direct excitation mode to 1:0.56 in cross-polarization mode (Figure 4, inset). The relative increase in the integral intensity for the peak at −32 ppm in ^1^H→^31^P CPMAS NMR spectrum may indicate a larger number of H atoms in the vicinity of the phosphorus atoms. These facts suggest the covalent functionalization of BP with CH_3_ groups with the formation of covalent P–CH_3_ and P(CH_3_)_2_ bonds in the BP–CH_3_ sample. Finally, by the integration of the signals in ^the 31^P MAS NMR spectrum, the functionalization degree of the BP–CH_3_ sample was roughly estimated: it was approximately 7.0% for P–CH_3_ fragments and 2.7% for P(CH_3_)_2_ fragments, with the overall functionalization degree of 9.7%.

X-ray photoelectron spectroscopy (XPS) was further used to characterize the BP and BP–CH_3_ samples. The P 2p core-level XPS spectrum of the BP sample (Figure 5a) exhibits characteristic peaks assigned to P 2p_1/2_ and P 2p_3/2_ at binding energies of 130.8 and 130.0 eV, respectively, as well as a broad low-intensity peak centered at 134.0 eV, assigned to the oxidized forms of phosphorus. The P 2p core-level XPS spectrum of BP–CH_3_ (Figure 5b) also contains high-intensity characteristic signals at 130.9 and 130.1 eV, which indicates the retention of the crystalline structure in the functionalized material. In addition, the spectrum exhibits a broadened signal in the region of 132.4–136.5 eV, which can be deconvoluted into three peaks centered at 133.7, 134.5 and 135.3 eV, which correspond to the P–C, P–O and P=O bonds, respectively. The signal in the high-resolution XPS spectrum of the C 1s core-level of the same sample (Figure 5c) can be deconvoluted into three peaks at 285.0, 285.9 and 286.6 eV, which are attributed to sp^2^/sp^3^ hybridized carbon atoms, C–O and C=O bonds, respectively. The obtained data also indicates the covalent functionalization of BP with methyl groups with the formation of the P–C bonds. 

The Raman spectrum of BP–CH_3_ recorded using 532 nm laser excitation (Figure 6) exhibits three characteristic signals at 358.7, 433.6 and 461.0 cm^−1^, corresponding to the A^1^_g_, B^2^_g_ and A^2^_g_ BP phonon modes, respectively. A slight blue shift of the A^1^_g_, B^2^_g_ and A^2^_g_ phonon modes in BP–CH_3_ compared to those signals in bulk BP located at 358.6, 432.5 and 460.1 cm^−1^, respectively, point out the ultrathin nature of the functionalized material [33,34]. For BP–CH_3_, in addition to the characteristic signals, new intensive peaks at 2906 and 2965 cm^−1^, along with a broadened peak in the region of 1265–1460 cm^−1^, are observed, which are assigned to different types of C–H vibrations [35]. These facts confirm the presence of methyl groups in the BP–CH_3_ sample. Moreover, the appearance of a new low-intensity peak at 705 cm^−1^ may be attributed to the P–C bond vibration [35].

Finally, we have implemented the method of Fourier-transform infrared spectroscopy (FT-IR) to characterize the functionalized material. Figure 7 displays the FT-IR spectra of BP and BP–CH_3_ samples. As a result of functionalization, the BP–CH_3_ sample exhibits peaks at 2992, 2910, 1405–1437, 1303 and 930–950 cm^−1^, which are assigned to different types of stretching and bending vibrations of the CH_3_ group. In addition, the spectrum also contains peaks in the region of 1010–1080 cm^−1^, corresponding to the stretching vibrations of the P–O bond, as well as signals at 694 and 667 cm^−1^, which belong to the asymmetric and symmetric vibrations of the P–C bond, respectively, and correlate well with the literature data [35,36].

Thus, the obtained spectroscopic data strongly suggest the covalent functionalization of BP with methyl groups during its cathodic exfoliation in the presence of Bu_4_NBF_4_ and CH_3_I. We assume that the functionalization proceeds as follows. At high negative potentials, the process of intercalation of tetrabutylammonium ions between the BP layers takes place, which is followed by their electrochemical reduction with the formation of gaseous products. The expansion of gaseous products, in turn, promotes the cleavage and exfoliation of BP [24]. Simultaneously with this process, the electrochemical reduction of CH_3_I also proceeds with the formation of a highly reactive methyl radical, which readily reacts with the material’s surface with the formation of P–C bonds. The use of BP as an electrode results in the effective functionalization of the material since, in this case, highly active free radicals are formed directly on the material’s surface, which greatly increases the probability of their successful interaction, bypassing side processes.

## 3. Materials and Methods

### 3.1. General Information

All manipulations and reactions were performed under an inert atmosphere of dry nitrogen using standard Schlenk-line or glovebox techniques. Organic solvents were distilled and stored under nitrogen before use. Commercially available reagent iodomethane (99%, Sigma-Aldrich, St. Louis, MO, USA) was used as purchased. The supporting electrolyte tetrabutylammonium tetrafluoroborate (99% Sigma-Aldrich, St. Louis, MO, USA) was melted in a vacuum to remove traces of residual water immediately before use. 

BP was obtained according to a modified procedure developed by Nilges [37]. The synthesis and characterization of BP have been published in our recent article [38]. In brief: 500 mg of red phosphorus, 364 mg of the Au/Sn alloy, and 10 mg of SnI_4_ were placed into a quartz ampoule 10 cm long and 10 mm in diameter (wall thickness 1 mm). The ampoule was sealed using a high-temperature burner at a pressure of 1 × 10^−6^ atm. Then the ampoule was placed into a two-zone tube furnace in such a way that the temperature of the empty side of the ampoule during synthesis was maintained at approximately 50 degrees lower than that of the sides with reagents. The ampoule with reagents was heated to 700 °C for 3 h. After that, the ampoule was kept at this temperature for 3 h, then slowly cooled to a temperature of 560 °C for 10 h. After cooling down, the ampoule was opened, and BP crystals formed at the empty side of the ampule were transferred into a flask with 10 mL of toluene. The mixture was refluxed for 1 h; after that, the precipitate was decanted, dried in vacuo, and stored under nitrogen. The product yield was 451 mg (90%), and purity was 99.6% (according to X-ray fluorescence spectroscopy analysis).

### 3.2. Characterization Techniques

Transmission electron microscopy (TEM) images of BP nanosheets were obtained on a Hitachi HT7700 transmission electron microscope (Tokyo, Japan) at an accelerating voltage of 100 kV. Samples were deposited on 300-mesh copper grids coated with lacey carbon support film (Electron Microscopy Sciences, Hatfield, PA, USA).

The surface morphology of BP nanosheets was studied by atomic force microscopy (AFM) in ambient conditions at tapping mode on a Titanium instrument (NT-MDT, Moscow, Russia) using a standard NSG-01 silicon cantilever (NT-MDT, Moscow Russia) with a resonant oscillation frequency of 120 kHz.

Solid-state NMR measurements were carried out on a Bruker AVANCE400 WB NMR spectrometer at a temperature of T = 293 K. The measurements were carried out on the ^31^P nuclei with a frequency of 162.056 MHz. For measurements, a MAS 4 BL CP BB DVT probe was used. The studied samples were packed into a 4 mm zirconium oxide rotor and were rotated at the magic angle at a speed of 12 kHz. The duration of the exciting 90-degree pulses for phosphorus nuclei was 3.4 µs. The standard impulse programs zg and hpdec, and cp were used. The number of points in the time domain was the same for all experiments and equal to 2048. The number of FID accumulations varied from 256 to 512 to obtain an acceptable signal-to-noise ratio. The time between FID accumulations in experiments on phosphorus nuclei was 20 s. The width of the spectra obtained for phosphorus nuclei was 69.5 kHz. In experiments with cross-polarization, the time varied from 1 to 5 ms to obtain the maximum signal.

X-ray photoelectron spectroscopy (XPS) measurements were carried out on a FleXPS spectrometer (SPECS, Berlin, Germany) with excitation by monochromatic Al Kα radiation. The pass energy of the electron energy analyzer was 20 eV. The vacuum in the system was 10^−9^ mbar. Powdered samples were applied to a double-sided adhesive copper foil. The element concentrations were calculated from the spectrum areas, taking into account the photoionization cross-section, the electron mean free path and the transmission function of the spectrometer. The binding energy of the C1s level from hydrocarbon surface contamination was 285.0 eV.

Raman spectra were collected with the Raman spectrometer SENTERRA (Bruker, Bremen, Germany). In order to register Raman spectra, linearly polarized light with a wavelength of 532 nm was used. Focusing of the laser beam on the surface of the sample was performed using a 100× objective with a numerical aperture of 0.9 (180° configuration). The laser power used in spectroscopic experiments was 2 mW. The exposure time was 60 s. Raman spectra were recorded within a spectral range of 45–3700 cm^−1^ with a resolution of 3–5 cm^−1^.

FT-IR spectra (500–4000 cm^−1^) were recorded using a Vertex 70 FT-IR spectrometer (Bruker, Bremen, Germany) equipped with an ATR accessory (MIRacle, PIKE Technologies, Madison, WI, USA). Background spectra obtained from 64 scans with a resolution of 2 cm^−1^ were subtracted from sample spectra.

In cyclic voltammetry (CV) studies, BP crystal and Pt electrodes with a working surface area of 2 mm^2^ were used as working electrodes. Voltammograms (CV curves) were recorded in a three-electrode electrochemical cell. CV curves were recorded in DMSO in the presence of 0.01 M Bu_4_NBF_4_ with a constant potential scan rate of 50 mV/s using BASI EC Epsilon potentiostat (West Lafayette, IN USA) equipped with a C3 cell stand. A silver electrode Ag/0.01 M AgNO_3_ solution in acetonitrile (E^0^(Fc/Fc+) = +0.20 V) served as a reference electrode. All potentials in this work are given relative to this reference electrode. A Pt wire 0.5 mm in diameter and 20 mm long was used as an auxiliary electrode. The measurements were carried out in a nitrogen atmosphere at room temperature.

### 3.3. BP–CH_3_ Synthesis

In an electrochemical cell with separation of the cathode and anode spaces, 25 mL of 0.01 M DMSO solution of Bu_4_NBF_4_ was added into each leg. 280 μL of CH_3_I (0.3 M) was added to the cathode part of the cell, after which a BP electrode was placed. A glassy carbon electrode was placed in the anodic part of the cell and served as an anode. Then the electrodes were connected to a power source, and the applied voltage was gradually increased from 1 to 5 V over 5 min. Next, the voltage was increased to 15 V and held for 10 min, after which the voltage was increased to 20 V and held for another 10 min. In the process of electrolysis, significant swelling of the BP electrode occurred, which was accompanied by the precipitation of the exfoliated BP into the solution. At the end of the electrosynthesis, the precipitate of functionalized BP nanosheets was transferred to a centrifuge tube and washed with DMSO and several portions of isopropanol by the successive centrifugation-redispersion procedures. The functionalized material was then transferred into a Schlenk flask and dried in vacuo.

## 4. Conclusions

In conclusion, we have developed a novel and facile approach for BP alkylation in mild conditions. Using iodomethane as an example, we have shown that an effective functionalization of BP may be achieved by its cathodic exfoliation in the presence of alkyl halide. We believe that the developed method may be expanded to provide the covalent functionalization of the material using alkyl halides with various compositions. Our study presents a significant step toward the preparation of new BP-based functional materials.

## Figures and Tables

**Figure 1 ijms-24-03095-f001:**
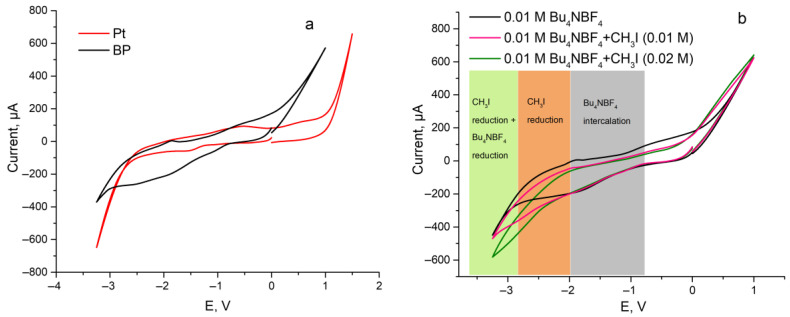
(**a**) CV-curves for a 0.01 M solution of Bu_4_NBF_4_ in DMSO; working electrodes: Pt and BP, v = 50 mV/s. (**b**) CV-curves for a 0.01 M solution of Bu_4_NBF_4_ in DMSO in the presence of the increasing amounts of CH_3_I; working electrode—BP, v = 50 mV/s.

**Figure 2 ijms-24-03095-f002:**
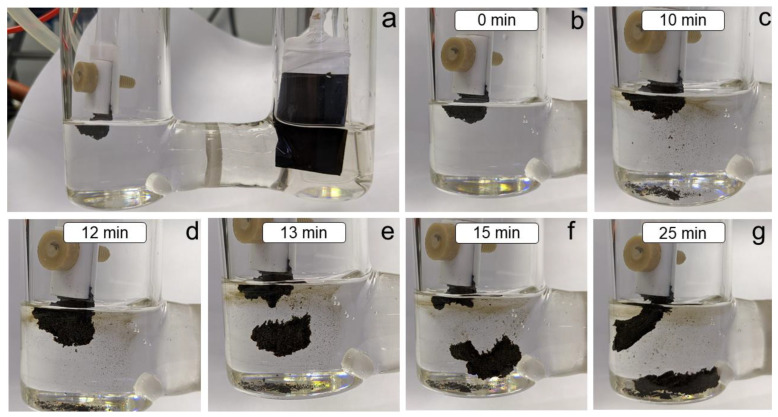
Photographs of the one-pot process of exfoliation and functionalization of BP. (**a**) A photograph of the electrochemical setup. (**b**–**g**) Photographs of the electrochemical exfoliation/functionalization process taken in 0, 10, 12, 13, 15 and 25 min after applied voltage, respectively.

**Figure 3 ijms-24-03095-f003:**
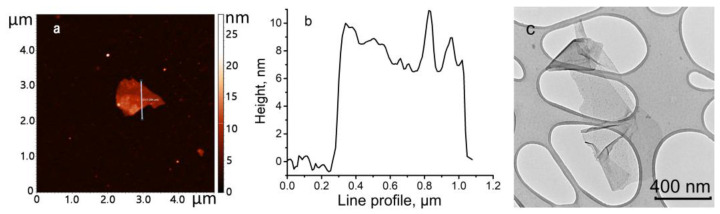
AFM image of BP–CH_3_ particle (**a**) and its line profile (**b**), as well as TEM image of BP–CH_3_ particles transferred on Lacey-carbon grid (**c**).

**Figure 4 ijms-24-03095-f004:**
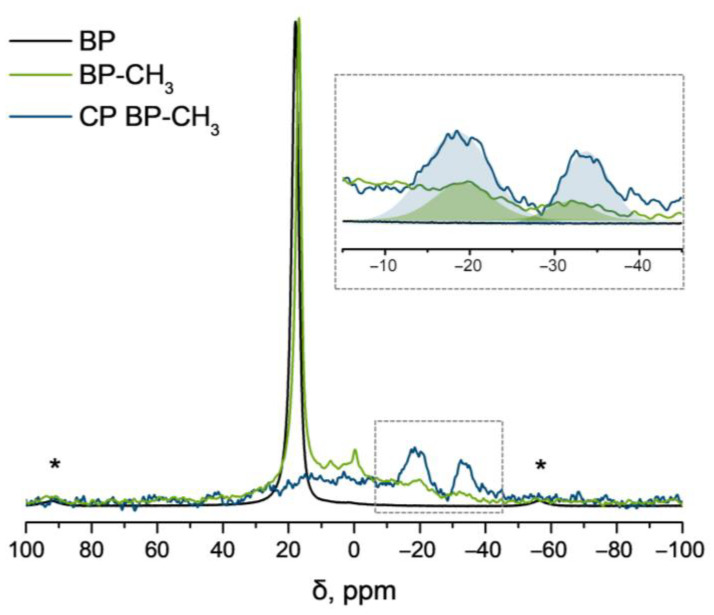
^31^P MAS NMR spectra of BP (black line) and BP–CH_3_ (green line) and ^31^P CPMAS NMR spectrum of BP–CH_3_ (blue line) (* denotes the peaks of spinning sideband). The inset contains the enlarged NMR spectra in the region of −5…−45 ppm; filled areas represent the deconvoluted peaks at −18 and −32 ppm in the corresponding NMR spectra of BP–CH_3_.

**Figure 5 ijms-24-03095-f005:**
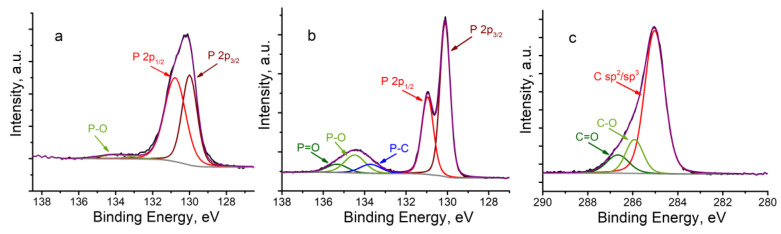
P 2p high-resolution XPS spectra of BP (**a**) and BP–CH_3_ (**b**) and C 1s high-resolution XPS spectrum of BP–CH_3_ (**c**). Experimental spectra, fitting lines and background lines are shown as black, purple, and gray lines, respectively.

**Figure 6 ijms-24-03095-f006:**
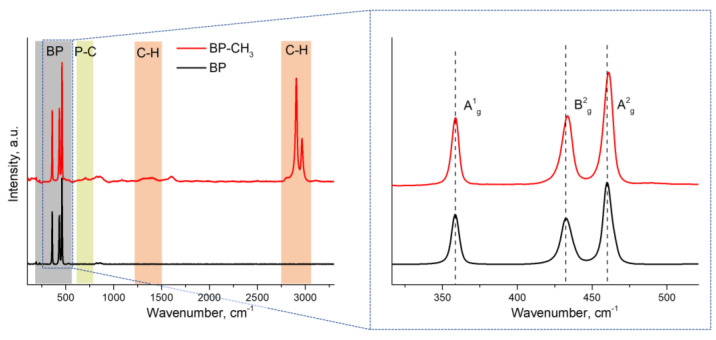
Raman spectra of BP and BP–CH_3_. The region of characteristic vibrations of BP is highlighted in gray, and the regions of P–C and C–H bond vibrations are highlighted in green and red, respectively. The inset contains the enlarged Raman spectra in the region of 315–520 cm^−1^; dashed lines represent the peak positions of A^1^_g_, B^2^_g_ and A^2^_g_ phonon modes in BP.

**Figure 7 ijms-24-03095-f007:**
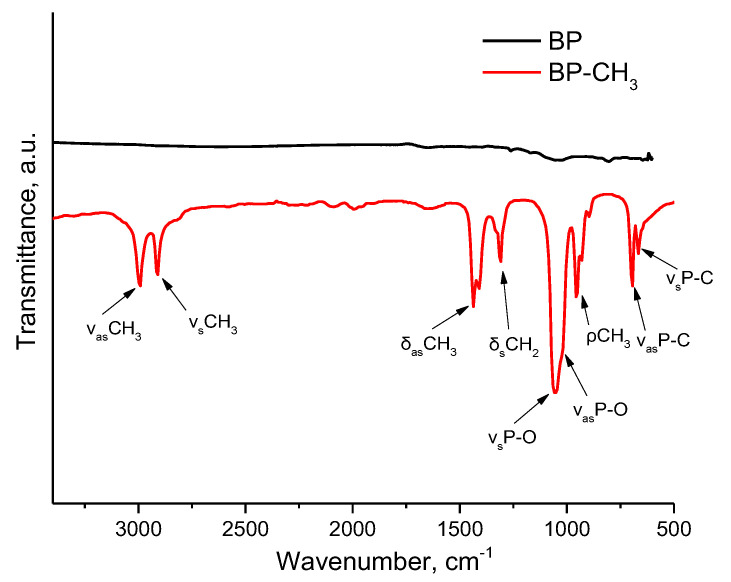
FT-IR spectra of BP and BP–CH_3_.

## Data Availability

The data presented in this study are contained within the article or are available upon request from the corresponding author, Dmitry G. Yakhvarov.

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
