# Peer review of "In-Situ Electrochemical Exfoliation and Methylation of Black Phosphorus into Functionalized Phosphorene Nanosheets"

_ijms, 2023, doi:10.3390/ijms24043095_

Round 1
Reviewer 1 Report
The author of paper which tile is “In situ electrochemical exfoliation and methylation of black phosphorus” evaluated the cathodic exfoliation and methylation of BP into phosphorene nanosheets. The paper is interesting, but it needs elaboration to make it more scientifically robust and significant. Please consult the following list for revision.
1- For the title, it is better to add the term of “into phosphorene” at the end after black phosphorus due to the universal term for the black phosphorus nanosheets which is phosphorene.
2- In the abstract part, authors need to provide some quantitative results to make the paper more attractive for the readers.
3- In the introduction part, for the first paragraph and introducing different properties of BP, authors cited somehow old references. I recommend checking the literatures in the field, for example they should cite the following paper which covered different types of preparation for black phosphorus: https://doi.org/10.1002/sstr.202000148
4- Providing more background of black phosphorus and its stability with or without additive is also necessary for this paper. Please highlight the stability of phosphorene nanosheets and then, explain the novelty of present work. For the stability of phosphorene nanosheets and its defect properties, authors can read the following paper and cite it as a reference:
https://doi.org/10.1016/j.susc.2022.122052
5- Although the authors cited their previous study for the synthesis of black phosphorus, but they need to highlight the method very briefly in the materials and methods part. What is the yield? Please provide a SEM/EDS image for the prepared black phosphorus to find out the quality of starting material sample.
6- From Figure 2, its clear that the chunk of bulk BP falling down in DMSO. What is the reason during the exfoliation process.
7- In terms of XPS analysis, please provide the XPS results for the bulk BP before exfoliation and compare it with the exfoliated phosphorene nanosheets.
8- Is it possible to mention the number of layers after exfoliation? Please check the layers using Raman analysis. Author can read the following paper in terms of TEM/Raman analysis of exfoliated phosphorene nanosheets and put it as a reference in the paper:
https://doi.org/10.1039/C9TA09641H
9- For the Raman analysis, its clear for the blue shift to the right, but why the intensity of peaks increased for the phosphorene nanosheets with CH3?
Author Response
Reviewer #1
The author of paper which tile is “In situ electrochemical exfoliation and methylation of black phosphorus” evaluated the cathodic exfoliation and methylation of BP into phosphorene nanosheets. The paper is interesting, but it needs elaboration to make it more scientifically robust and significant. Please consult the following list for revision.
- For the title, it is better to add the term of “into phosphorene” at the end after black phosphorus due to the universal term for the black phosphorus nanosheets which is phosphorene.
Response: The title is changed to “In situ electrochemical exfoliation and methylation of black phosphorus into functionalized phosphorene nanosheets”
- In the abstract part, authors need to provide some quantitative results to make the paper more attractive for the readers.
Response: The functionalization degree estimated by NMR analysis was added to the abstract part.
- In the introduction part, for the first paragraph and introducing different properties of BP, authors cited somehow old references. I recommend checking the literatures in the field, for example they should cite the following paper which covered different types of preparation for black phosphorus: https://doi.org/10.1002/sstr.202000148
Response: We are thankful to the Reviewer for this remark. The corresponding article has been cited in the introduction part. The BP synthesis and different methods for its exfoliation are also covered in the cited reviews [1, 3, 9, 10].
- Providing more background of black phosphorus and its stability with or without additive is also necessary for this paper. Please highlight the stability of phosphorene nanosheets and then, explain the novelty of present work. For the stability of phosphorene nanosheets and its defect properties, authors can read the following paper and cite it as a reference: https://doi.org/10.1016/j.susc.2022.122052
Response: Thank you for this remark! The mentioned article has been cited in the paper. The environmental stability of BP nanosheets is mentioned in the introduction part. Unfortunately, we cannot perform the special stability test due to a short period of time given for the revisions. However, the oxidation stability of methylated black phosphorus was provided by Hirsch et. al. in https://doi.org/10.1002/anie.201811181. The novelty of the present research is focused on the development of a new efficient method for BP alkylation, rather than the preparation of new BP-based materials with enhanced ambient stability. The developed alkylation method makes it possible to avoid the use of BP intercalation compounds in the synthetic procedures.
- Although the authors cited their previous study for the synthesis of black phosphorus, but they need to highlight the method very briefly in the materials and methods part. What is the yield? Please provide a SEM/EDS image for the prepared black phosphorus to find out the quality of starting material sample.
Response: The description of the procedure for the black phosphorus preparation has been added to the experimental section of the manuscript. The BP samples were characterized by single crystal X-ray diffraction, X-ray powder diffraction and X-ray fluorescence spectroscopy analysis. The yield was about 90%, purity was 99,6 % (according to X-ray fluorescence spectroscopy analysis).
- From Figure 2, its clear that the chunk of bulk BP falling down in DMSO. What is the reason during the exfoliation process.
Response: The detached parts of the BP electrode were actually the exfoliated BP nanosheets. The swelling and expansion of the electrode is due to the formation of gaseous compounds during the cathodic reduction of tetrabutylammonium salt intercalated between BP layers. With a high degree of exfoliation, individual parts of the BP crystal are not able to hold the exfoliated BP, which leads to its transition into solution. The process of exfoliation of BP is especially noticeable in the figures 2f and 2g, where a large volume of exfoliated material is formed from a very small area of the electrode contacting the electrolyte solution. Similar effects of exfoliated BP transferring into solution are also observed in the literature [https://doi.org/10.1039/C9CC07640A, https://doi.org/10.1021/acs.chemmater.8b00521, https://doi.org/10.1002/ange.201705071]. The absence of bulk BP in the obtained material can be indirectly proven by Raman spectroscopy, where only signals belonging to the thin layers are observed.
- In terms of XPS analysis, please provide the XPS results for the bulk BP before exfoliation and compare it with the exfoliated phosphorene nanosheets.
Response: The corresponding spectra (XPS spectra for bulk BP and the comparison of BP and BP-CH3) were added to the paper.
- Is it possible to mention the number of layers after exfoliation? Please check the layers using Raman analysis. Author can read the following paper in terms of TEM/Raman analysis of exfoliated phosphorene nanosheets and put it as a reference in the paper: https://doi.org/10.1039/C9TA09641H
Response: We thank the Reviewer for the suggestion. The mentioned article was cited in the paper. The blue shift in Raman spectra indicates the thin nature of the obtained BP nanosheets. The number of the layers after the exfoliation was estimated by AFM analysis (according to the obtained data, there are 14 – 18 layers in the sample). We couldn’t use the A2g/A1g intensities ratio for determining the material’s thickness, since according to the literature [https://doi.org/10.1007/s12274-014-0446-7, https://doi.org/10.1038/nmat4299] it is only applicable to determine the thickness of BP nanosheets containing 5 or less phosphorene monolayers. Moreover, these signals are also quite susceptible to the oxidation, so the use of a direct method for determining the particle thickness such as AFM seems more convenient.
- For the Raman analysis, its clear for the blue shift to the right, but why the intensity of peaks increased for the phosphorene nanosheets with CH3?
Response: The Raman spectra of BP and BP-CH3 given in the paper are not quantitative. For the sample preparation an adequate amount of suspension of BP or BP-CH3 in isopropanol were drop casted to a thin glass plate and dried in vacuo. The higher intensities of BP-CH3 signals are due to slightly bigger amounts of the sample registered by Raman spectrometer. Usually for the quantitative analysis, the signals of the samples are normalized according to the external signal of Si plate. In our case, there were no signals of the glass plate suitable for the normalization.

Reviewer 2 Report
The paper is focused on electrochemical exfoliation and methylation of black phosphorous. The topic falls within the scope of the journal. I recommend the publication after the following revisions.
- Figure 3. The scale length within AFM and TEM images is not clear. Please check and revise.
- I suggest to determine the sizes distribution of the BP–CH3 particles from a statistical analysis of TEM images.
- It would be interesting to compare the morphological characteristics of BP-CH3 with those of BP.
- Lines 165-168. The authors stated “For BP–CH3, in addition to the characteristic signals, new intensive peaks at 2906 and 2965 165 cm–1 along with a broadened peak in the region of 1265–1460 cm–1 are observed, which are assigned to different types of C–H vibrations. These facts confirm the presence of methyl groups in the BP–CH3 sample.” This statement should be supported by related literature.
- It could be added a Conclusions paragraph.
- The novelty and interest of this paper should be better highlighted in the Abstract.
Author Response
Reviewer #2
The paper is focused on electrochemical exfoliation and methylation of black phosphorous. The topic falls within the scope of the journal. I recommend the publication after the following revisions.
- Figure 3. The scale length within AFM and TEM images is not clear. Please check and revise.
Response: The images are corrected for improved readability
- I suggest to determine the sizes distribution of the BP–CH3 particles from a statistical analysis of TEM images.
Response: Unfortunately, we cannot perform the size distribution analysis using statistical TEM images due to too few numbers of particles captured by the microscope. Additionally, the stacking of the nanosheets also complicates the analysis.
- It would be interesting to compare the morphological characteristics of BP-CH3 with those of BP.
Response: The morphology description of the obtained BP-CH3 nanosheets were added to the main body text of the manuscript. The functionalized material retains its crystallinity, which was further proven by Raman, XPS, and solid-state NMR spectroscopy methods.
- Lines 165-168. The authors stated “For BP–CH3, in addition to the characteristic signals, new intensive peaks at 2906 and 2965 165 cm–1 along with a broadened peak in the region of 1265–1460 cm–1 are observed, which are assigned to different types of C–H vibrations. These facts confirm the presence of methyl groups in the BP–CH3” This statement should be supported by related literature.
Response: The corresponding reference has been added.
- It could be added a Conclusions paragraph.
Response: The “Conclusions” paragraph has been added.
- The novelty and interest of this paper should be better highlighted in the Abstract.
Response: The Abstract was corrected according to the Reviewer’s remark and the novelty of the performed study has been better highlighted.

Round 2
Reviewer 1 Report
The paper is ready to publish in a present form.